# The Role of MicroRNA in the Pathogenesis of Atopic Dermatitis

**DOI:** 10.3390/ijms26125846

**Published:** 2025-06-18

**Authors:** Natalia Gołuchowska, Aldona Ząber, Agata Będzichowska, Agata Tomaszewska, Agnieszka Rustecka, Bolesław Kalicki

**Affiliations:** 1Department of Pediatrics, Pediatric Nephrology and Allergology, Military Institute of Medicine—National Research Institute, 04-141 Warsaw, Poland; azaber@wim.mil.pl (A.Z.); awawrzyniak@wim.mil.pl (A.T.); arustecka@wim.mil.pl (A.R.); kalicki@wim.mil.pl (B.K.); 2Faculty of Medicine, University of Warsaw, 02-089 Warsaw, Poland

**Keywords:** atopic dermatitis, miRNA, Th2-dependent inflammation, treatment, biomarkers

## Abstract

Atopic dermatitis (AD) is a chronic inflammatory skin disease characterized by eczematous lesions and recurrent pruritus. The pathogenesis of AD involves epidermal barrier impairments, immune dysfunction, and both genetic and environmental factors. This review presents the latest findings regarding the involvement of microRNA (miRNA) molecules in AD pathogenesis and their potential application as diagnostic biomarkers and therapeutic targets. The following miRNAs were analyzed in detail: miR-223, miR-10a-5p, miR-29b, miR-146a-5p, miR-451a, miR-124, miR-143, miR-151a, miR-24, miR-191, and miR-155. Their molecular mechanisms and potential clinical implications are discussed. Understanding the role of specific microRNAs in AD pathogenesis may contribute to the development of new diagnostic and therapeutic strategies for this disease.

## 1. Introduction

Atopic dermatitis (AD) is a chronic, noninfectious inflammatory skin disease. It is characterized by recurrent, intense itching and eczematous lesions [1,2]. The incidence of AD is increasing, particularly in developed countries, affecting 15–30% of children and 2–10% of adults worldwide [3]. AD is a chronic condition with recurrent flare-ups, and approximately 50–80% of patients also have other atopic diseases [4]. A study published in Pediatric Allergy and Immunology showed that food allergies occur in 15–30% of children with moderate to severe AD [5]. Moreover, approximately 30% of pediatric patients with AD develop asthma later in life [6]. Allergic rhinitis occurs in approximately 50–75% of patients with AD [7].

The pathogenesis of AD has not been fully elucidated [8]. It is known to be associated with epidermal barrier abnormalities (including filaggrin structure and function, tight junctions, and lipid composition), immune system dysregulation, and neurovegetative disorders. All these factors contribute to maintaining a chronic inflammatory state [3]. Th2-dependent inflammation plays a key role in the pathogenesis of atopic dermatitis (AD). In AD, epidermal barrier dysfunction associated with mutations in the filaggrin gene serves as a starting point for the activation of dendritic cells and initiation of the Th2 response [4]. Activated Th2 lymphocytes produce a characteristic cytokine profile, including interleukin 4 (IL-4), interleukin 13 (IL-13), and interleukin 31 (IL-31), which not only stimulate IgE production but also directly affect keratinocytes, disrupting the expression of structural proteins in the epidermis and exacerbating skin barrier dysfunction [9]. Furthermore, Th2 cytokines induce the expression of chemokines such as CCL17 and CCL22, which recruit additional T cells to the skin, creating a positive feedback loop [4]. In AD, increased activity of type 2 innate lymphoid cells (ILC2) is also observed, which, like Th2 lymphocytes, produce interleukin 5 (IL-5) and IL-13, contributing to tissue eosinophilia and chronic inflammation [10,11]. Microbiome imbalances are also a significant feature of this Th2-mediated disease [12,13,14]. A damaged epidermal barrier (reduced lipid content, alkaline pH, decreased levels of antibacterial peptides—defensins and cathelicidin) promotes skin colonization by bacterial strains (including *Staphylococcus aureus, Staphylococcus epidermidis*) and fungi (including *Candida albicans* and *Malassezia* spp.), which contributes to the exacerbation of skin inflammation [15,16,17,18]. Additionally, the skin neuro–immuno–endocrine system communicates with the local microbiome, neural, endocrine and immune systems through the production of soluble factors, priming circulating immune cells or neural transmission. [19,20]. The AD manifests in genetically predisposed individuals under the influence of environmental and personal factors, such as food allergens, airborne allergens, infections, irritants, and stress [21].

Predisposition to the disease is polygenically inherited and involves genes encoding epidermal proteins (filaggrin gene FLG, genes encoding serine protease inhibitors SPINK-5/LEKTI1, cystatin A, and others), proteins involved in innate and acquired immune responses, and genes for neuropeptide receptors and genes regulating vitamin D synthesis and action [21,22,23].

MicroRNAs (miRNAs) are short, single-stranded, non-coding RNA molecules consisting of approximately 19–25 nucleotides, first identified in 1993 [24,25,26].

MiRNAs are transcribed from DNA in the cell nucleus, forming primary miRNA transcripts (pri-miRNAs), which are then processed into precursor miRNAs (pre-miRNAs) and transported to the cytoplasm [24,25,26]. In the cytoplasm, pre-miRNAs undergo further processing into mature miRNAs, which are incorporated into the RNA-induced silencing complex (RISC). The miRNA–RISC complex recognizes a complementary sequence on the target mRNA, consisting of 7–8 nucleotides, causing inhibition of translation or degradation of the attached mRNA, thereby regulating gene expression at the post-transcriptional level. This regulatory mechanism is essential for maintaining normal cell function and development [24,25] [Figure 1].

Negative regulation of gene expression by microRNAs plays an important role in homeostasis. It is reported that approximately one-third of gene expression pathways involve microRNA. It should also be noted that one type of microRNA can control approximately two hundred genes. One specific gene, however, can be regulated by multiple microRNAs. These molecules ultimately ensure the proper amount of individual proteins by participating in the degradation of transcripts that are in excess or are no longer needed. Disturbed miRNA expression thus negatively affects the protein balance in the cell [25,27].

In clinical practice, biomarkers serve as a helpful tool in diagnostics and treatment monitoring because they provide objective and consistent results compared to using scales and physical examination [28].

The levels of characteristic biomarkers in blood plasma/serum are often measured to detect disease, predict its course, or monitor therapeutic effects. For this reason, increasing attention is being paid to circulating microRNAs, whose profiles, characteristic of a given disease entity, are highly specific and allow confirmation of the presence of the disease and prediction of its development with high sensitivity [29,30,31,32].

One of the advantages of using circulating miRNAs as clinical markers is the minimal invasiveness and ease of obtaining biological material for testing. Another important feature enabling the use of miRNA molecules as diagnostic and prognostic indicators is their significant quantity and availability in body fluids and the possibility of studying expression using traditional molecular biology tools [29,30,31,32,33].

Changes in miRNA expression have been documented in numerous disease entities, including inflammatory and immunological dermatological conditions, contributing to a better understanding of their pathogenesis and the development of new biomarkers and therapeutic strategies, e.g., in cancers, neurodegenerative diseases, diabetes, cardiovascular diseases, or hepatitis C [29,33,34,35,36,37].

In this review, we present a detailed characterization of individual miRNAs that may play a key role in the pathogenesis of AD and discuss their potential use as diagnostic biomarkers and therapeutic targets.

## 2. Materials and Methods

This literature review aimed to assess the current understanding of microRNA’s involvement in the pathogenesis of atopic dermatitis. The following electronic databases were searched: PubMed, Scopus, Web of Science, and Google Scholar. A combination of keywords and Boolean operators was used to identify relevant articles. The primary search terms included ‘Atopic Dermatitis’, ‘eczema’, ‘miRNA’, ‘expression’, and ‘biomarkers’. The inclusion criteria were: articles published in peer-reviewed journals and studies involving human populations. The exclusion criteria were: non-original articles, publications unrelated to atopic dermatitis or miRNA. All searches were conducted twice to ensure comprehensive coverage of the topic. Additionally, titles and abstracts of the results were manually reviewed to identify the most relevant studies in the context of microRNA expression in atopic dermatitis.

## 3. MicroRNAs

### 3.1. MicroRNA-223

Research suggests that microRNA-223 (miR-223) plays a key role in the pathogenesis of AD. A relationship is suggested between this molecule and changes in the fetal immune system, leading to increased susceptibility to AD in children exposed to tobacco smoke in the prenatal period [38,39]. Exposure to tobacco smoke in utero can affect T-regulatory cells, with miR-223 likely acting as a molecular mediator of this effect [40,41].

MiR-223 exerts a suppressive effect on the differentiation and function of Treg cells, which are essential for inhibiting the excessive immune response observed in the pathogenesis of AD [39]. Herbert et al. demonstrated that in children with prenatal exposure to tobacco smoke, there is an elevated concentration of miR-223 in serum, which may lead to an imbalance resulting in heightened immune activation and the development of inflammation, potentially contributing to increased susceptibility to AD [38,41,42].

The literature also provides reports of a correlation between miR-223 concentration and the severity of AD [43]. Yasuike et al. conducted a study involving 21 adult patients assigned to the following groups: patients with AD (divided into mild, moderate, and severe forms according to the Eczema Area and Severity Index—EASI scale), patients with urticaria, and healthy volunteers as a control group. Patients received standard treatment, including topical glucocorticosteroids, calcineurin inhibitors, and systemic antihistamines. Individuals taking systemic glucocorticosteroids or immunosuppressive drugs were excluded from the study. A statistically significant increase in miR-223 expression in plasma was observed in patients with severe AD compared to patients with urticaria and healthy volunteers. Moreover, plasma miR-223 expression levels in patients with severe AD were significantly higher than in patients with mild and moderate AD. These results suggest that plasma miR-223 may serve as a potential biomarker for AD severity [43]. However, it is important to note that the study involved a very small patient group, which represents a major limitation.

### 3.2. MicroRNA-10a-5p

MicroRNA-10a-5p (miR-10a-5p) plays a notable role in the pathogenesis of AD. Its dysregulation has significant implications for susceptibility to atopic dermatitis. MicroRNA-10a-5p has been identified as a regulator of keratinocyte proliferation [44]. Keratinocytes are the dominant cell type in the epidermis, forming the outermost layer of the skin. Dysregulation of miR-10a-5p can disrupt the delicate balance of keratinocyte dynamics, impairing skin barrier function [44]. Vaher et al. demonstrated significantly increased expression of miR-10a-5p in both diseased skin and seemingly unaffected skin of patients with AD compared to a control group [44].

In vitro studies revealed that miR-10a-5p inhibits keratinocyte proliferation through direct interaction with hyaluronic acid synthase 3 (HAS3) [44]. HAS3 is a positive regulator of keratinocyte proliferation and migration associated with damage. A damaged barrier allows allergens and irritants to penetrate the skin more easily, triggering immune reactions characteristic of AD [45,46]. Additionally, the influence of miR-10a-5p on the expression of genes involved in cell cycle regulation, cell adhesion, and cytokine signaling pathways, including mitogen-activated protein kinase kinase kinase 7 (MAP3K7), was observed [44]. Modulation of cytokine expression, such as interleukin 1β (IL-1β), IL-4, interleukin 8 (IL-8), interleukin 17A (IL-17A) and CCL5, was also demonstrated, although its impact on inflammatory processes is less pronounced than its effect on keratinocyte proliferation. Furthermore, elevated miR-10a-5p expression was associated with higher levels of the Ki-67 proliferation marker in the skin of patients with AD, suggesting a complex role for miR-10a-5p in immune regulation and the pathogenesis of AD [44].

### 3.3. MicroRNA-29b

When analyzing the relationship between miRNA and AD, attention should also be paid to microRNA-29b (miR-29b), which plays an important role in regulating keratinocyte apoptosis and maintaining epithelial barrier integrity [47]. It directly interacts with BCL-2-like protein 2 and mediates IFN-γ-related apoptosis in keratinocytes. By targeting specific genes involved in apoptosis, miR-29b ensures that the epidermis maintains an optimal balance between cell death and cell proliferation [47,48].

Gu et al. provided evidence for the involvement of miR-29b in the pathogenesis of AD [49]. Their study included two groups: 21 patients with AD and 12 healthy individuals as a control group. Skin biopsies were collected from the study participants. Additionally, serum samples from 60 individuals (30 with AD and 30 healthy controls) were analyzed serologically. Gu et al. found significantly elevated levels of miR-29b in both skin lesions and blood serum of patients with AD compared to the control group. Moreover, they observed a positive correlation between miR-29b concentration and the severity of AD, assessed using the SCORAD (Scoring Atopic Dermatitis) scale, suggesting a potential role of miR-29b as a biomarker. In vitro studies on keratinocytes demonstrated that miR-29b enhances IFN-γ-induced apoptosis by inhibiting BCL2L2 (BCL2-like 2) expression. The research suggests that the miR-29b/BCL2L2 regulatory axis plays a significant role in the pathogenesis of AD, modulating miR-29b levels may represent a potential therapeutic strategy [49].

### 3.4. MicroRNA-146a-5p

MicroRNA-146a-5p (miR-146a-5p) also plays an important role in the pathogenesis of AD by regulating immune system function, particularly through the NF-κB (nuclear factor kappa B) signaling pathway, and is significantly correlated with IgE levels in AD [50,51].

In the course of AD, the NF-κB pathway often shows excessive activity, leading to an enhanced immune response and characteristic inflammatory symptoms. MiR-146a-5p acts as a negative regulator of NF-κB signaling by targeting key components of the pathway, such as interleukin-1 receptor-associated kinase 1 (IRAK1), tumor necrosis factor receptor-associated factor 6 (TRAF6), and caspase recruitment domain-containing protein 10 (CARD-10) [50,51].

Studies by Rebane et al. showed that miR-146a-5p also interacts with C-C motif chemokine ligand 5 (CCL5) [50]. MiR-146a-5p downregulates these proteins, thereby dampening NF-κB activation, which leads to decreased production of proinflammatory cytokines and chemokines, mitigating the inflammatory cascade observed in AD [52].

Research in mice provides strong evidence that T cells lacking miR-146a lead to increased Th1/Th17-type immune responses [52], where overexpression of miR-146a at the organismal level induces a class switch in B cells and overproduction of IgE [53]. The AD phenotype is more severe in miR-146a-deficient mice, without affecting type 2 cell-mediated immune responses in the skin [54]. Carreras-Badosa et al. analyzed the relationship between miR-146a-5p and IgE production in mice and in patients with AD [54]. Their study showed that mice with miR-146a-5p deficiency exhibited reduced serum IgE levels and elevated IL-12p40 (interleukin 12 subunit p40) levels compared to wild-type mice. These results suggested that miR-146a-5p plays a role in IgE production in mice. No significant differences in miR-146a and IL-12p40 levels were found in patients with AD. In the next stage of analysis, the researchers assessed the relationship between miR-146a-5p and the form of AD. The researchers did not observe differences in serum miR-146a levels between allergic and non-allergic patients. However, it was observed that serum miR-146a levels were negatively associated with serum IgE levels in patients with the allergic subtype of AD. These findings suggest that in the allergic subtype of AD, miR-146a-5p may function as an inhibitory regulator of type 2 immune reactions [54].

### 3.5. MicroRNA-451a

The literature also reports a correlation between the pathogenesis of AD and the expression of microRNA-451a (miR-451a). Nousbeck et al. conducted a comprehensive analysis of miRNA expression profiles in peripheral blood from infants with AD compared to a control group of healthy infants. Their study included 100 infants with AD, and the control group consisted of 20 healthy children [55]. RNA sequencing analysis showed significant differences in miRNA expression in peripheral blood mononuclear cells (PBMCs) and plasma. In PBMCs from infants with AD, increased expression of miRNAs associated with inflammatory processes was observed, including miR-223-3p, miR-143-3p, and miR-126-5p, as well as decreased expression of miR-451a. In plasma, dysregulation of eight miRNAs (miR-1290, miR-451a, miR-501-3p, miR-193a-3p, miR-885-3p, miR-3652, miR-4447, miR-1287-5p) was observed. Dysregulation of three miRNAs, miR-451a, miR-143-3p and miR-223-3p was validated in a larger number of samples. Notably, miR-451a was the only miRNA dysregulated in both PBMCs and plasma. Furthermore, the authors confirmed that interleukin 6 receptor (IL6R) and proteasome subunit beta type 8 (PSMB8), which are targets of miR-451a, showed increased expression in patients with AD and a negative correlation with miR-451a concentration. The study emphasizes the importance of miR-451a, miR-223, and miR-143-3p in the pathogenesis of AD and their potential application in the early diagnosis of this inflammatory dermatosis [55].

### 3.6. MicroRNA-124

Previous studies have shown a significant role of microRNA-124 (miR-124) in inflammatory processes. One main mechanism through which miR-124 exerts its anti-inflammatory effect is direct targeting and inhibition of components of the NF-κB pathway [56,57,58].

By reducing NF-κB activity, miR-124 effectively suppresses proinflammatory cytokine expression, thereby decreasing the recruitment and activation of immune cells, mitigating the inflammatory response observed in atopic dermatitis [56,57,58].

Yang et al. demonstrated that p65 mRNA (an NF-κB subunit), interleukin 8 (IL-8), and chemokines CCL5 and CCL8 were increased in affected skin areas of patients with AD. Their expression inversely correlated with miR-124 levels, which were decreased [59]. The researchers concluded that miR-124 controls NF-κB-dependent inflammatory responses in keratinocytes and chronic skin inflammation in AD, suggesting that restoring miR-124 expression may represent a promising therapeutic strategy for AD [59].

### 3.7. MicroRNA-143

The microRNA-143 (miR-143) molecule is one of the key regulatory elements of Th2-dependent inflammation through its influence on the IL-13R1 receptor. IL-13, secreted mainly by activated Th2 cells and mast cells, is believed to play a role in the development of allergic inflammation in atopic dermatitis [60].

Increased IL-13 expression is associated with weakened expression of epidermal barrier proteins, including filaggrin (FLG), loricrin (LOR), and involucrin (IVL), leading to abnormal permeability homeostasis and adversely affecting epidermal barrier function [61,62,63]. The mechanism through which IL-13 inhibits FLG, LOR, and IVL expression in human epidermal keratinocytes remains unclear. A study conducted by Zeng et al. showed that the expression of miR-143 is reduced in human keratinocytes exposed to IL-13. In the study, miR-143 inhibited IL-13 activity and inflammatory processes through interaction with IL-13Rα1 in keratinocytes, resulting in increased integrity of the epidermal barrier [60,64].

### 3.8. MicroRNA-151a

Studies by Chen et al. demonstrated the involvement of microRNA-151a (miR-151a) in the pathogenesis of AD [65]. The scientists examined serum samples from 500 patients with AD and 200 healthy volunteers serving as controls. Among patients with AD, significantly elevated levels of miR-151a in blood serum were observed. The researchers showed that miR-151a inhibits the expression of interleukin-12 receptor β2 (IL12RB2), thereby reducing the production of cytokines characteristic of Th1 lymphocytes. Inhibition of Th1 activity may in turn disrupt the Th1/Th2 balance and lead to the development of Th2-dependent inflammation [65].

### 3.9. MicroRNA-24 and MicroRNA-191

According to studies by Meno et al., the expression of microRNA-24 (miR-24) and microRNA-191 (miR-191) plays a significant role in the pathogenesis of AD [66]. In their work, the researchers demonstrated elevated levels of miR-24 and miR-191 in the blood plasma of patients with severe AD compared to patients with mild AD and urticaria and healthy volunteers. Furthermore, the concentrations of these miRNAs positively correlated with the levels of TARC (Thymus and Activation-Regulated Chemokine), also known as CCL17—C-C motif chemokine ligand 17, a chemoattractant for Th2 cells in blood serum [66]. It is well known that the level of TARC in patients with AD is elevated both in blood serum and in skin biopsies. Moreover, the level of TARC in serum closely correlates with the severity of the disease in patients with AD [66,67]. A correlation was also observed between the concentrations of miR-24 and miR-191 in blood plasma and the levels of PF-4 (platelet factor 4) and β-TG (beta-thromboglobulin), indicators of platelet activation, which contribute to the exacerbation of AD [67]. PF-4, belonging to the CXC chemokine family, after release from the α granules of activated platelets, exhibits strong chemotactic properties towards neutrophils and T lymphocytes, intensifying their influx into skin lesions in AD [68,69]. Studies have shown that platelet-derived PF-4 induces monocyte survival through the inhibition of apoptosis and promotes monocyte differentiation into macrophages which can be associated with the development of chronic inflammation in atopic skin [69,70,71].

Beta-thromboglobulin, also released from the α granules of platelets during their activation, promotes fibroblast chemotaxis and stimulates their proliferation, potentially contributing to tissue remodeling and lichenification characteristic of chronic AD. Studies have shown significantly elevated levels of PF-4 and β-TG in the serum of patients with AD compared to the control group, correlating with the severity of clinical symptoms, suggesting their potential role as biomarkers of disease activity [66,72,73].

### 3.10. MicroRNA-155

MicroRNA-155 (miR-155) plays a key role in regulating the immune response and the differentiation of Th17 (type 17 helper T) cells, making it a promising diagnostic and therapeutic target in AD [74,75,76].

Additionally, miR-155 directly targets CTLA-4 (cytotoxic T-lymphocyte-associated protein 4), a negative regulator of T cell activation [75]. In AD, miR-155 plays a multifaceted role in modulating the immune response, influencing the differentiation and function of immune cells, particularly T lymphocytes and dendritic cells [74,75]. MiR-155 is essential for the proliferation of Th17 cells and their cytokines, such as interleukin-17 (IL-17), which play an important role in the pathogenesis of AD. MiR-155 promotes Th17 cell differentiation by interacting with key regulatory genes, indicating its involvement in immune system dysregulation in AD [74,77]. Disturbances in the Th17 response contribute to chronic inflammation and tissue damage in skin affected by AD [74].

The significance of miR-155 in AD extends beyond immune regulation to include diagnostic and therapeutic potential. Altered miR-155 expression in patients with AD suggests that elevated levels in skin samples or peripheral blood may serve as diagnostic markers, especially in difficult-to-diagnose cases [77].

The impact of miR-155 on Th2 cells is complex and remains unclear. The literature lacks conclusive evidence on whether miR-155 always acts as a suppressor or activator of the Th2 response [78,79].

In a study by Zitner et al., inhibition of miR-155 expression promoted the Th2 response and increased the secretion of IL-4 and interleukin-10 (IL-10) [80]. In another study, inhibition of miR-155 resulted in increased expression of GATA3 (GATA binding protein 3), which is a key transcription factor for Th2 [81]. It is also suggested that miR-155 may inhibit Th2 cell differentiation through repression of the c-Maf transcription factor [82].

The literature also indicates that increased miR-155 expression stimulates the production of Th2 cytokines, such as IL-5 and IL-13 [83]. In studies on mice with reduced miR-155 expression, decreased levels of IL-4, IL-5, and IL-13 were observed, which may be related to the influence of miR-155 on the PU.1 transcription factor [84,85]. These discrepancies highlight the complexity of Th2 response regulation by miR-155 and the need for further research to fully explain the mechanisms of its action.

## 4. Conclusions

Atopic dermatitis is a chronic inflammatory dermatosis with a complex pathogenesis. This review presents the current state of knowledge regarding the involvement of selected miRNAs in the pathogenesis of AD, considering their potential application as diagnostic biomarkers and therapeutic targets. Analysis of the available scientific literature indicates a significant role of several miRNAs in the development of AD: miR-223, miR-10a-5p, miR-29b, miR-146a-5p, miR-451a, miR-124, miR-143, miR-151a, miR-24, miR-191, and miR-155 [Table 1 and Figure 2]. Dysregulation of miRNAs is linked to aberrant skin barrier function, cytokine signaling and NF-κB-dependent inflammatory responses, together with Th17, Th1, Th2, Treg and platelet activities. However, the roles of many miRNAs associated with AD are still not well understood. More comprehensive studies are required to better evaluate the involvement of these miRNAs in AD development. MiRNA-regulated biological processes offer important therapeutic opportunities. For example, there is growing interest in treatments aimed at restoring the skin barrier in AD research [86]. Inhibition of NF-κB has been demonstrated to decrease disease severity in AD mouse models [87]. Additionally, therapies that target Th17 have shown promising results in recent clinical trials for AD [88]. The expansion of Treg cells in a therapeutic context could potentially reduce the allergic inflammatory response in AD [89]. These new perspectives indicate promising avenues for innovative AD drug development. The literature contains numerous examples that highlight the potential of miRNAs as novel diagnostic biomarkers. However, findings from miRNA research in AD often lack conclusive evidence due to the small and varied sample sizes typically used in these studies.

It is necessary to continue research to deepen knowledge about the correlation between the expression of individual types of miRNAs and AD, considering their role in the immune response. Future research should focus on validating potential miRNA biomarkers in large patient cohorts for reliable diagnostics. Longitudinal studies are needed to assess miRNAs’ predictive power for AD onset and progression. Development of miRNA-based therapeutics, such as mimics or inhibitors, shows promise for targeted treatment of AD. Finally, examining miRNAs’ role in different types of AD may refine personalized therapeutic strategies, ultimately optimizing patient outcomes.

## Figures and Tables

**Figure 1 ijms-26-05846-f001:**
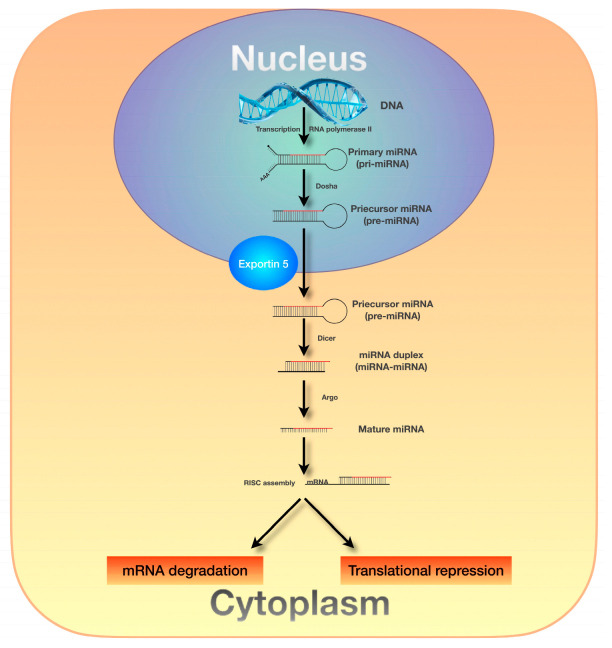
MiRNA biogenesis pathway.

**Figure 2 ijms-26-05846-f002:**
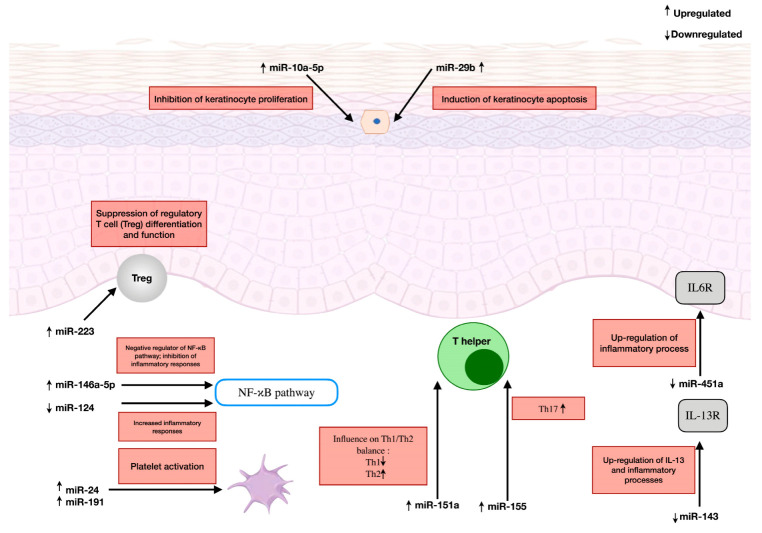
The role of microRNA in the pathogenesis of atopic dermatitis.

**Table 1 ijms-26-05846-t001:** The role of microRNA in the pathogenesis of atopic dermatitis.

MicroRNA	Expression in AD	Mechanism of Action	Molecular Targets	Potential Clinical Significance	Experimental Model of Molecular Targets
miR-223	Upregulated	Suppression of regulatory T cell (Treg) differentiation and function	Not precisely determined	Biomarker of AD severity; associated with prenatal tobacco smoke exposure	Human
miR-10a-5p	Upregulated	Inhibition of keratinocyte proliferation; modulation of cytokine expression (IL-1β, IL-4, IL-8, IL-17A, CCL5)	Hyaluronic acid synthase 3 (HAS3); MAP3K7 kinase	Regulator of skin barrier integrity	In vitro
miR-29b	Upregulated	Regulation of IFN-γ-induced keratinocyte apoptosis	BCL2L2 (BCL2-like 2)	Biomarker correlating with AD severity (SCORAD)	In vitro
miR-146a-5p	Upregulated	Negative regulator of NF-κB pathway; inhibition of inflammatory responses	IRAK1, TRAF6, CARD-10, CCL5	Potential regulator inhibiting type 2 immune responses	Human
miR-151a	Upregulated	Influence on Th1/Th2 balance	Interleukin-12 receptor β2 (IL12RB2)	Potential AD biomarker	In vitro
miR-24	Upregulated	Association with platelet activation	Likely associated with TARC (CCL17)	Biomarker of AD severity; correlation with platelet activation	Human
miR-191	Upregulated	Association with platelet activation	Likely associated with TARC (CCL17)	Biomarker of AD severity; correlation with platelet activation	Human
miR-155	Upregulated	Regulation of immune response; influence on Th17cell differentiation	CTLA-4; transcription factor c-Maf; transcription factor PU.1	Potential diagnostic marker; complex effect on Th2 cells	Animal (mice)
miR-451a	Downregulated	Regulation of inflammatory processes	Interleukin 6 receptor (IL6R); proteasome subunit beta type 8 (PSMB8)	Potential diagnostic marker of AD in infants	Human
miR-124	Downregulated	Control of NF-κB-dependent inflammatory responses	p65 subunit (NF-κB)	Potential therapeutic target	In vitro
miR-143	Downregulated	Inhibition of IL-13 activity and inflammatory processes	Interleukin-13 receptor alpha 1 (IL-13Rα1)	Regulator of inflammatory cascade in AD	In vitro

## Data Availability

The datasets used during the current case report are available from the corresponding author upon reasonable request.

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
