# Peer review of "The Role of MicroRNA in the Pathogenesis of Atopic Dermatitis"

_ijms, 2025, doi:10.3390/ijms26125846_

Round 1

Reviewer 1 Report

Comments and Suggestions for Authors

The manuscript provides a well-structured and comprehensive review of the role of microRNAs in the pathogenesis of atopic dermatitis. The authors effectively summarize current findings and elucidate the molecular mechanisms through which specific miRNAs contribute to immune dysregulation and skin barrier dysfunction. The topic is timely and relevant, and the review is well-supported by current literature.

I recommend minor revisions to enhance scientific language, eliminate redundancy, and improve overall clarity. Additionally, expanding the final table with an extra column indicating the experimental model (e.g., human, animal, in vitro) would strengthen its clarity and utility.

Overall, this is a valuable contribution to the field and is suitable for publication following minor revisions.

Author Response

Comments 1: I recommend minor revisions to enhance scientific language, eliminate redundancy, and improve overall clarity. Additionally, expanding the final table with an extra column indicating the experimental model (e.g., human, animal, in vitro) would strengthen its clarity and utility.

Response 1: I sincerely thank you for dedicating your valuable time and expertise to review my manuscript. Your insightful feedback and comments have been immensely helpful in refining the quality of my work. I greatly appreciate your contributions, which have significantly enhanced the overall impact of the study. The corrections related to your comment can be found in Table 1. The manuscript has been thoroughly revised and carefully proofread to improve the language and ensure clarity and correctness throughout. We believe that the overall readability and quality of the text have been improved.

Reviewer 2 Report

Comments and Suggestions for Authors

Very interesting and important content.

Please provide the basis for the order in which the various miRNA are presented.

The information in Table 1 could be more effectively presented with a composite graphical backdrop of various cellular components involved in the different mechanisms of action.

Comments on the Quality of English Language

The manuscript requires proofreading and editing by a native speaker of English as the expression is often unidiomatic throughout.

Author Response

Comments 1: Please provide the basis for the order in which the various miRNA are presented.

Response 1: I am grateful for the time and effort you invested in evaluating my manuscript. Your constructive comments and suggestions have been very valuable in improving the clarity and rigor of my research. Thank you for your valuable input, which has greatly contributed to the development of my work. The order in which the various miRNAs are presented is random and does not reflect any specific ranking or categorization. We chose this approach for simplicity and readability.

Comments 2: The information in Table 1 could be more effectively presented with a composite graphical backdrop of various cellular components involved in the different mechanisms of action.

Response 2: We agree with this comment, therefore we have prepared the suggested graphic (Figure 2., page 10, line 376). 

The manuscript has been thoroughly revised and carefully proofread to improve the language and ensure clarity and correctness throughout by native speaker. We believe that the overall readability and quality of the text have been improved.

Reviewer 3 Report

Comments and Suggestions for Authors

This review article describes the role of microRNAs in the pathogenesis of atopic dermatitis.

The article is interesting and well written.

I would suggest that the authors add a search strategy to explain how they comprehensively searched Medline/PubMed and other databases.

The authors should discuss in more detail the role of microRNAs in the diagnosis of atopic dermatitis, as well as in monitoring therapy. Could microRNAs be a therapeutic target in atopic dermatitis?

I would suggest that in the Conclusions section the authors discuss the perspectives for further research on the role of microRNAs in atopic dermatitis.

Author Response

 Comments 1: I would suggest that the authors add a search strategy to explain how they comprehensively searched Medline/PubMed and other databases.

Response 1: Thank you for taking the time to carefully review my submission. Your helpful insights and recommendations have played a significant role in enhancing the quality of my work. I truly appreciate your effort and the constructive feedback provided. We agree with this comment. You can find the revised fragment at „Materials and methods” ( page 3, line 103-113).

Comments 2: The authors should discuss in more detail the role of microRNAs in the diagnosis of atopic dermatitis, as well as in monitoring therapy. Could microRNAs be a therapeutic target in atopic dermatitis?

Response 2: We agree with this comment. You can find the revised fragment at conclusions (page 8, line 349-372).

Comments 3: I would suggest that in the Conclusions section the authors discuss the perspectives for further research on the role of microRNAs in atopic dermatitis.

Response 3: Thank You for pointing this out. You can find the revised fragment at conclusions (page 8, line 349-372).